# Genomic and Biochemical Characterization of *Bifidobacterium pseudocatenulatum* JCLA3 Isolated from Human Intestine

**DOI:** 10.3390/microorganisms10112100

**Published:** 2022-10-22

**Authors:** Raquel González-Vázquez, Eduardo Zúñiga-León, Edgar Torres-Maravilla, Martha Leyte-Lugo, Felipe Mendoza-Pérez, Natalia C. Hernández-Delgado, Ricardo Pérez-Pastén-Borja, Alejandro Azaola-Espinosa, Lino Mayorga-Reyes

**Affiliations:** 1Laboratorio de Biotecnología, Departamento de Sistemas Biológicos, Unidad Xochimilco, CONACYT-Universidad Autónoma Metropolitana, Ciudad de Mexico 1100, Mexico; 2Laboratorio de Biotecnología, Departamento de Sistemas Biológicos, Unidad Xochimilco, Universidad Autónoma Metropolitana, Ciudad de Mexico 1100, Mexico; 3INRAE, AgroPArisTEch, Micalis Institute, Université Paris-Saclay, 78350 Jouy-en Josas, France; 4Laboratorio de Toxicología Molecular y Celular, Escuela Nacional de Ciencias Biológicas-Campus Zacatenco, Instituto Politécnico Nacional, Ciudad de Mexico 07738, Mexico

**Keywords:** *Bifidobacterium pseudocatenulatum*, gastrointestinal tract stress, potential probiotic characteristics, genome sequencing

## Abstract

Bifidobacteria have been investigated due to their mutualistic microbe–host interaction with humans throughout their life. This work aims to make a biochemical and genomic characterization of *Bifidobacterium pseudocatenulatum* JCLA3. By multilocus analysis, the species of *B. pseudocatenulatum* JCLA3 was established as *pseudocatenulatum*. It contains one circular genome of 2,369,863 bp with G + C content of 56.6%, no plasmids, 1937 CDSs, 54 tRNAs, 16 rRNAs, 1 tmRNA, 1 CRISPR region, and 401 operons predicted, including a CRISPR-Cas operon; it encodes an extensive number of enzymes, which allows it to utilize different carbohydrates. The *ack* gene was found as part of an operon formed by *xfp* and *pta* genes. Two genes of *ldh* were found at different positions. Chromosomally encoded resistance to ampicillin and cephalothin, non-hemolytic activity, and moderate inhibition of *Escherichia coli* ATCC 25922 and *Staphylococcus aureus* ATCC 6538 were demonstrated by *B. pseudocatenulatum* JCLA3; it can survive 100% in simulated saliva, can tolerate primary and secondary glyco- or tauro-conjugated bile salts but not in a mix of bile; the strain did not survive at pH 1.5–5. The *cbh* gene coding to choloylglycine hydrolase was identified in its genome, which could be related to the ability to deconjugate secondary bile salts. Intact cells showed twice as much antioxidant activity than debris. *B. pseudocatenulatum* JCLA3 showed 49% of adhesion to Caco-2 cells. The genome and biochemical analysis help to elucidate further possible biotechnological applications of *B. pseudocatenulatum* JCLA3.

## 1. Introduction

Bifidobacteria are part of the first bacteria that colonize the gut shortly after birth and establish a mutualistic microbe–host interaction, influencing early and late life [1,2]. Many species of the genus *Bifidobacterium* are currently being studied for their potential health benefits and their effects on metabolic disorders associated with obesity by modulating gut microbiota composition and promoting changes in lipid metabolism and glucose homeostasis [3], by their ability to modulate the production of gamma amino butyric acid in the human gastrointestinal tract, by acting as a defense against pathogens, by hydrolyzing non-digestible dietary carbohydrates through the expression of enzymes and producing short-chain fatty acids (SCFA), mainly lactate, acetate and, formate, in addition to being able to stimulate the immune response [1,3,4,5,6].

There is an evident lack of knowledge concerning the molecular mechanisms that explain these probiotic traits of *Bifidobacterium*. The complete genome sequencing of *Bifidobacterium* may be helpful to better predict more accurate functional properties for health benefits and clarify the mechanisms involved in its interaction with the host, mainly how they tolerate gastrointestinal tract stress [7]. The European Food Safety Authority has developed guidelines for the safety assessment of probiotics, which include a taxonomic identification of the strain by whole-genome sequencing, genomic and phenotypic determination of the antibiotic resistance profile, and safety evaluation [8].

Particularly, *B. pseudocatenulatum* has shown probiotic properties, such as the possession of antinutrient-degrading enzymes, the ability to bind mutagenic aromatic amines, the capacity to reduce cholesterol levels [9], clinical applications of aging-related diseases and extending life span [10]. Olvera et al. (2013) [11] isolated a strain of *Bifidobacterium* from a natural newborn, who was breastfeeding; it was identified only at the genera level, and the expression of *ack* and *ldh* genes were studied under different substrates. In this work, we determined that the *Bifidobacterium* isolated by Olvera et al. (2013) [11] belongs to the species *pseudocatenulatum*. The knowledge of the complete genome of this *B. pseudocatenulatum* would provide greater insight into the taxonomic affiliation and intra-specific variation of this species and supply information on the genetics that underlay strain-specific capabilities as have been reported in other *Bifidobacterium* [9]. Thus, this study aimed to create a genomic and biochemical characterization of the previously isolated *B. pseudocatenulatum* JCLA3 to further understand the role of this bacterium in the metabolism of carbohydrates and the catalytic tools it uses to adapt to the gastrointestinal tract of mammals, its antibiotic susceptibility, its tolerance to the gastrointestinal tract (GIT), and adhesion properties. It also aims to provide valuable information on its potential functional properties, such as antimicrobial, bile salt hydrolase (BSH), and antioxidant activity.

## 2. Materials and Methods

### 2.1. Isolation

The bacteria isolation from a newborn and identification at the genera level of *B. pseudocatenulatum* JCLA3 were previously reported by Olvera et al., 2013 [11].

### 2.2. DNA Extraction and Genotypic Identification

According to the manufacturer’s instructions, the total gDNA was isolated using the Wizard^®^ Genomic DNA Purification Kit (Promega). The 16S rDNA was sequenced at the Divisional Molecular Biology Laboratory of Universidad Autónoma Metropolitana campus Iztapalapa, México City, using the set of primers Bif 164 and Bif 662 [12]. Comparisons and sequence alignments were made using MEGA5 [13] and NCBI´s primary local alignment search tool (http://blast.ncbi.nlm.nih.gov/Blast.cgi; accessed on 9 May 2022).

### 2.3. Multilocus Sequence Analysis (MLSA)

This method is based on sequence comparison of housekeeping genes in the bacterial genome and generated a robust and highly discriminatory super tree that has been used to infer phylogeny in the genus *Bifidobacterium* [14]. MLSA was performed by concatenating fragments from seven conserved genes, i.e., *clpC*, *dnaB, dnaG*, *dnaJ1*, *purF*, *rpoC*, and *xfp*. The homologous sequences of the target genes were obtained from 12 bifidobacterial strains’ genomes using an in-house python script, concatenated and aligned with Clustal W [15]. Phylogenetic relationships were inferred using Maximum Likelihood-based on the Kimura 2-parameters model [16]. Bootstrap values of the tree were computed by resampling 100 times. The phylogenetic analysis was conducted in MEGA7 [17].

### 2.4. Genome Sequencing and Annotation

The samples were sequenced at the Integrated Microbiome Resource (IMR, Dalhousie University, Halifax, NS, Canada) [18]. The assembly was also conducted by IMR using SMRT analysis software. After assembling the whole genome, the location of protein-coding sequences, tRNA genes, rRNA genes, tmRNA genes, and CRISPR were predicted using Prokka v1.12-beta [19]. Prokka conducts analysis based on ISfinder, NCBI Background Reference Gene DB (with tbl2asn v25.8 and blastp v2.2 tool), UniProtKB DB (with blastp v2.2 tool), and HMM DB (with hmmscan v3.1 tool). The following tools were used to predict each functional element: Prodigal v2.6 [20] for CDS prediction, RNAmmer v1.2 [21] for rRNA prediction, Aragorn v1.2 [22] for tRNA and transfer-messenger RNA (tmRNA) prediction, MinCED (https://github.com/ctSkennerton/minced; accessed on 13 June 2022) and CRISPRFinder [23] (last modified 9 May 2017) for Clustered regularly interspaced short palindromic repeats (CRISPRs) prediction and cmscan v1.1 [24] for sRNA prediction.

The protein-coding genes were functionally annotated using an hmm search (HMMER v3.3.2, November 2020) [25] with HMMs profiles (for a total of 4983 profiles) of Bifidobacteriales (last modified on 23 March 2020) extracted from the eggNOG v5.0.0 database [26]. We opted for HMMs profiles to obtain homologous sequences because of their sensitivity to detecting homology. The data were processed in a Python environment v3.6.7 through the platform Jupyter Notebook v6.0.3 using Pandas v1.0.4 and Numpy v1.18.5 packages. The Classes of Clusters of orthologous groups (COG) and Functional Categories assignment were performed to an E-value cutoff of 1E-5.

Enzymatic classification in *B. pseudocatenulatum* JCLA3 was predicted from *B. pseudocatenulatum* JCM 1200 deposited on the KEGG database (GenBank Assembly GCA_001025215.1, T_number: T03932 and Org_code: bpsc). The assignment was BLAST All-versus-All based using Blastp v2.8.1 and an in-house python script. The data were processed in a Python environment v3.6.7 through the platform Jupyter Notebook v6.0.3 using Pandas v1.0.4 and Numpy v1.18.5 packages. The best hits were selected to an E-value cutoff of 1E-5 and an Identity percentage greater than 40%. In addition, the architecture between both sets of proteins was compared using the Pfam functional annotation. The Pfam domains were predicted using pfam_scan.pl script v1.6. A circular plot was used to visualize CDS sequences and Enzyme classes. The circular plot was performed using an in-house python script through the platform Jupyter Notebook v6.0.3 using functions of the Matplotlib v3.0.3 package.

The putative operons for all CDSs were predicted using the Operon-Mapper tool [27], where default settings were used. GFF and FNA (complete genome in FASTA format) files were used. The data were processed through Jupyter Notebook v6.0.3 using Pandas v1.0.4 and Numpy v1.18.5 packages. A circular plot was used to visualize CDS sequences and Enzyme classes. The circular plot was performed as mentioned above. For the visualization of the genome, the following information was used: (i) the sequence of the complete genome in FASTA format (*.fna), and (ii) the genomic features in GFF (General Feature Format) format v3.0. The Genome circular plot was performed as mentioned above. The functional annotation and their distribution were also included in the circular plot.

### 2.5. Effect of Carbon Source on Growth of B. pseudocatenulatum JCLA3

The growth kinetics was followed during 8 h of fermentation at 37 °C, using TPY medium independently added with 1% *w*/*v* of glucose (as a control), lactose, sucrose, or inulin. All media were adjusted to pH 7 with NaOH 0.1 M (Thermo Scientific Orion 410A+, MA, USA), and oxygen was removed from the medium by bubbling CO_2_ and sealed with a rubber stopper before sterilization. The samples were taken from time zero to 8 h every 2 h. The increase in biomass (mg/mL) was determined through optical density and a standardized curve of dry weight. The biomass was separated by centrifuging for 5 min at 3000× *g*. The pH of the supernatant was measured and then discarded. The pellet was resuspended in 1 mL of water, and the optical density was determined at 660 nm (Varian Inc., Cary 50, Palo Alto, CA, USA). All experiments were conducted in triplicate and are expressed as a mean with standard deviation.

### 2.6. Antibiotic Profile

The ability to resist antibiotics was determined using the multidisc PT-34 Multibac I.D. (Investigación Diagnóstica, CdMx, México) following the instructions of the supplier, which included β-lactam antibiotics: vancomycin (30 μg), ampicillin (10 μg), dicloxacillin (1 μg), cephalothin (30 μg), penicillin (10 U), cefotaxime (30 μg); inhibitors of protein synthesis gentamicin (10 μg), clindamycin (30 μg), erythromycin (15 μg), tetracycline (30 μg); inhibitors of nucleic acid synthesis ciprofloxacin (5 μg), and others, such as trimethoprim-sulfamethoxazole (25 μg). The profile of *B. animalis* subsp. *lactis* Bb-12 (CHR Hansen) was determined to compare the antibiotic profile of *B. pseudocatenulatum* JCLA3. All the experiments were carried out in triplicate [28].

### 2.7. Hemolysis Test

Before the test, the strains were incubated overnight in MRS broth supplemented with sucrose (2 g/L) and cysteine (0.5%). A 1 × 10^8^ CFU/mL sample was plated onto blood agar and incubated for 48 h at 37 °C in an anaerobic chamber (Forma anaerobic system Model 1025, Thermo Scientific, Pittsburgh, PA, USA) with an atmosphere of 10% CO_2_, 5% H_2,_ and 85% N_2_. For each strain, the test was carried out in triplicate. *Escherichia coli* ATCC 160211 was used as a positive control, and *B. animalis* subsp. *Lactis* Bb-12 as a negative control. The experiments were conducted in triplicate. Results were reported as follows: alpha indicating partial hemolysis, beta indicating total hemolysis, and gamma as non-hemolytic [28].

### 2.8. Antimicrobial Activity

The agar well diffusion assay [29] was used to evaluate the ability of the strain to inhibit *E. coli* ATCC 25922, *E. coli* O157:H7, *Salmonella typhi* ATCC14028, and *Staphylococcus aureus* ATCC 6538. *B. pseudocatenulatum* JCLA3 and *B. animalis* subsp. *lactis* Bb-12 (control) were grown on TPY agar at 1 × 10^7^ CFU/mL concentration for each one and incubated at 37 °C for 24 h. Afterward, that warm, soft tryptic soy agar (8 g/L) was emptied into each Bifidobacterium culture. Once gelled, the other bacteria were plated individually onto the agar at a concentration of 10^8^ CFU/mL. The plates were incubated at 37 °C for 24 h in an anaerobic atmosphere. The experiments were conducted in triplicate. The results were expressed as follows: no inhibition (-), moderate (+), and total inhibition (++).

### 2.9. Bile Salt Tolerance Assay

BSH activity was qualitatively determined by plate assay. Fresh bacterial cultures were dropped onto TPY agar containing 0.1, 0.2, 0.3, and 0.5% (*w*/*v*) glycocholic or taurocholic acid (Sigma Aldrich, USA). Bacterial cultures were then anaerobically incubated at 37 °C for 48 h in an anaerobic atmosphere. Strains with BSH activity were surrounded by a halo of precipitated deconjugated bile salts [30] and were considered positive. *B. animalis* subsp. *lactis* Bb-12 was used as a positive control, and the experiments were conducted in triplicate.

### 2.10. Tolerance to the Gastrointestinal Tract (GIT)

This tolerance implied the ability to survive simulated saliva (SS), simulated gastric fluid (SGF), and simulated intestinal fluid (SIF). SS consisted of a sterilized solution of pH 6.2 of phosphate buffer (PBS) and 0.01% (*w*/*v*) lysozyme. SGF was prepared with PBS adjusted with HCL (1M) (J.T. Baker, USA) to pH 1.5 and 0.1% pepsin. All the simulated fluids were prepared according to [28,31]. The SIF consisted of a sterilized PBS pH 6.8 supplemented with 0.5% (*w*/*v*) oxgall and 0.1% pancreatin. The tolerance was determined by comparing the cell count before and after exposure to each stress condition. The initial concentration of 1 × 10^8^ CFU /mL was mixed with SS and incubated for 10 min at 37 °C under anaerobic conditions with gentle shaking. After incubation, the solution was separated by centrifugation at 3000× *g* for 5 min; the pellet was washed twice with phosphate buffer pH 7, suspended in 1 mL SGF, and incubated at 37 °C for 90 min in anaerobic conditions with gentle shaking. Afterward, the solution was centrifuged under the same conditions mentioned above to separate the pellet, then suspended in 1 mL SIF and incubated at 37 °C for 150 min with gentle shaking. The experiments also were conducted using the same solutions with or without enzymes and in a sequential and non-sequential manner. The viability was also tested at pH 2, 3, and 5. *B. animalis* subsp. *lactis* Bb-12 was tested to compare the experiments. All the experiments were carried out in triplicate. The results were expressed as % of viability, considering the initial count by plate assay as 100%.

### 2.11. DPPH Radical Scavenging Activity Assay

The 2,2-diphenyl-1-picrylhydrazyl (DPPH, Sigma) radical scavenging activity to test antioxidant activity in intact cells, intracellular cell-free extracts, and cell debris of *B. pseudocatenulatum* JCLA3 and *B. animalis* subsp. *lactis* Bb-12 was conducted according to the method proposed by Su et al. (2015) [32]. The strains were harvested by centrifugation at 4400× *g* for 10 min. The supernatant was considered a cell-free extract. For the preparation of intact cells, the pellet was washed three times with PBS and finally suspended in the same buffer. For the preparation of cell debris, overnight cultures were treated as above, and cell extract suspended in PBS was subjected to ultrasonic disruption (1:1 min treatment: ice bath, 5 times) and then centrifuged at 7800× *g* for 20 min at 4 °C, the supernatant was discarded, and the cell debris was suspended in PBS.

The DPPH assay of the strains was analyzed by mixing one milliliter of freshly prepared DPPH solution (0.2 mmol/L in methanol) with 1 mL of sample solution. Then, the mixture was allowed to react for 30 min in the dark. The absorbance was measured with a UV-visible spectrophotometer at 517 nm. The assay was performed by three independent experiments. The DPPH scavenging ability was calculated using the formula:$$\%DPPH=1-\left[Abs_{517}\left(sample\right)/Abs_{517}\left(PBS\right)\right]\times 100$$

### 2.12. Adhesion Assay

Caco-2 cells (ATCC^®^ HTB-37™) were cultivated in Eagle’s minimal essential medium (Gibco Invitrogen, Carlsbad, CA, USA) supplemented with 20% fetal bovine serum (FBS; Gibco Invitrogen, Carlsbad, CA, USA), 1% penicillin/streptomycin, 1% L-glutamine, and 1% (*v*/*v*) nonessential amino acids solution (Gibco). The cells were seeded in 24-well culture plates (1  ×  10^5^ cells per well) and incubated at 37 °C in 5% CO_2_. After about 24 h, a confluent monolayer was obtained, and the plates were incubated for 15 days. Overnight growth of the strain *B. pseudocatenulatum* JCLA3 or *B. animalis* subsp. *lactis* Bb-12 (used to establish the 100% adhesion) was harvested by centrifugation at 6000× *g* for 5 min at 4 °C, washed twice with PBS (pH 7.4), and suspended in Dulbecco’s Modified Eagle Medium without antibiotic and fetal bovine serum. Each bacterial suspension (1 mL of 1 × 10^8^ CFU/mL) was added to the 24-well plates and incubated for 1 h at 37 °C in a 5% CO_2_ atmosphere. After, each well was washed three times with PBS (pH 7.4) to remove non-adherent bacteria. The adhered bacteria were detached from the monolayer with a solution of trypsin–EDTA (Gibco) in PBS; the enzymes were inactivated with a culture medium. Serial dilutions were plated onto MRS agar plates supplemented with 0.5% L-cysteine hydrochloride (Merck), and the bacterial colonies were counted after 48 h of incubation under anaerobic conditions [33]. All the experiments were carried out in triplicate.

### 2.13. Statistical Analysis

Media and standard deviation were calculated in all the experiments. The results of the experiments were compared by T student or ANOVA using *p* < 0.05.

## 3. Results

### 3.1. Genotypic Identification

The multilocus analysis showed that JCLA3 belongs to the *pseudocatenulatum* species (Figure 1). The complete genome was registered at Gene bank CP090598 as *B. pseudocatenulatum* JCLA3. The BioProject accession number is PRJNA795522.

### 3.2. Genome Analysis

*B. pseudocatenulatum* JCLA3 has one circular genome of 2,369,863 bp with a G + C content of 56.6%, with no plasmids (Figure 2). The genomic annotations illustrated a total of 1937 coding sequences (CDSs).

The genome of *B. pseudocatenulatum* JCLA3 possessing 54 tRNAs, 16 rRNAs (including 16S, 23S, and 5S rRNA), 1 tmRNA, and 1832 CDSs (94.57%) were assigned to COG categories. In addition, the *B. pseudocatenulatum* JCLA3 contained a clustered regularly interspaced short palindromic repeats (CRISPR) region (1,723,829 to 1,726,407) associated with *cas* genes.

The 23.46% of the COG category assignment were of genes with unknown function, 9.79% carbohydrate transport and metabolism, 8.92% amino acid transport and metabolism, 7.34% translation ribosomal structure, and biogenesis, 6.83% transcription and 6.83% replication, recombination, and repair, whereas 5.43% genes were not placed into the COGs (Figure 2).

The genome of *B. pseudocatenulatum* JCLA3 also encodes an extensive number of enzymes: 184 transferases (11.28%), 178 hydrolases (10.91%), 67 oxidoreductases (4.11%), 64 ligases (3.92%), 47 lyases (2.88%), 38 isomerases (2.33%) and 21 translocases (1.29%) (Figure 3 and Appendix A).

*B. pseudocatenulatum* JCLA3 contain two *ldh* genes, one at position 62,018 to 63,001 and the other at 1,593,405 to 1,594,367 bp; and one *ack* gene (968,782 to 970,011 bp), encoding a lactate dehydrogenase (EC1.1.1.27) and acetate kinase (EC 2.7.2.1), respectively. The ack gene was found as part of an operon formed by three genes (*xfp*, *pta* and *ack*) at positions 964,348 to 970,011 bp (Figure 3) that correspond to -xylulose 5-phosphate/d-fructose 6-phosphate phosphoketolase (EC 4.1.2.9), phosphate acetyltransferase (EC 2.3.1.8), and acetate kinase (EC 2.7.2.1).

In addition, the *B. pseudocatenulatum* JCLA3 genome contains 401 operons with more than two CDS, some shown in Figure 3. Moreover, the genome showed biosynthetic capabilities regarding amino acids, such as arginine, valine, leucine, isoleucine, lysine, histidine, phenylalanine, tyrosine, and tryptophane. According to the genome information, the biosynthesis of arginine could be mediated by glutamate dehydrogenase (EC 1.4.1.4), aspartate aminotransferase (EC 2.6.1.1), argininosuccinate lyase (EC 4.3.2.1), argininosuccinate synthase (EC 6.3.4.5), ornithine carbamoyltransferase (EC 2.1.3.3), acetylornithine/N-succinyldiaminopimelate aminotransferase (EC 2.6.1.11 and 2.6.1.17), acetylglutamate kinase (EC 2.7.2.8), glutamate N-acetyltransferase (EC 2.3.1.35), amino-acid N-acetyltransferase (EC 2.3.1.1), N-acetyl-gamma-glutamyl-phosphate reductase (EC 1.2.1.38), glutamine synthetase (EC 6.3.1.2) and N-acetylglutamate synthase (EC 2.3.1.1). Valine, leucine and isoleucine biosynthesis could involve ketol-acid reductoisomerase (EC 1.1.1.86), 2-isopropylmalate synthase (EC 2.3.3.13), 3-isopropylmalate/(R)-2-methylmalate dehydratase (EC 4.2.1.33 and 4.2.1.35), acetolactate synthase I/III (EC 2.2.1.6), dihydroxy-acid dehydratase (EC 4.2.1.9), branched-chain amino acid aminotransferase (EC 2.6.1.42), alanine-synthesizing transaminase (EC 2.6.1.66), and 3-isopropylmalate dehydrogenase (EC 1.1.1.85). Lysine could be synthesized by aspartate kinase (EC 2.7.2.4) and aspartate-semialdehyde dehydrogenase (EC 1.2.1.11), succinyl-diaminopimelate desuccinylase (EC 3.5.1.18), 2,3,4,5-tetrahydropyridine-2,6-dicarboxylate N-succinyltransferase (EC 2.3.1.117), acetylornithine/N-succinyldiaminopimelate aminotransferase (EC 2.6.1.11 and 2.6.1.17), UDP-N-acetylmuramoyl-L-alanyl-D-glutamate-2,6-diaminopimelate ligase (EC 6.3.2.13), UDP-N-acetylmuramoyl-tripeptide-D-alanyl-D-alanine ligase (EC 6.3.2.10), diaminopimelate epimerase (EC 5.1.1.7), 4-hydroxy-tetrahydrodipicolinate synthase (EC 4.3.3.7), 4-hydroxy-tetrahydrodipicolinate reductase (EC 1.17.1.8), homoserine dehydrogenase (EC 1.1.1.3), and diaminopimelate decarboxylase (EC 4.1.1.20). The biosynthesis of histidine is carried out by 1-(5-phosphoribosyl)-5-[(5-phosphoribosylamino) methylideneamino] imidazole-4-carboxamide isomerase (EC 5.3.1.16 and 5.3.1.24). Phenylalanine, tyrosine, and tryptophane could be biosynthesized by aspartate aminotransferase (EC 2.6.1.1), 3-deoxy-7-phosphoheptulonate synthase (EC 2.5.1.54); 3-phosphoshikimate 1-carboxyvinyltransferase (EC 2.5.1.19), chorismate synthase (EC 4.2.3.5), shikimate kinase/3-dehydroquinate synthase (EC 2.7.1.71), 3-dehydroquinate dehydratase II (EC4.2.1.10), anthranilate synthase component I (EC 4.1.3.27), anthranilate phosphoribosyltransferase (EC 2.4.2.18), prephenate dehydrogenase (EC 1.3.1.12), chorismate mutase (EC 5.4.99.5) prephenate dehydratase (EC 4.2.1.51), phosphoribosyl isomerase A (EC 5.3.1.16), and histidinol-phosphate aminotransferase (EC 2.6.1.9).

*B. pseudocatenulatum* JCLA3 showed genes involved in the biosynthesis of hydrosoluble vitamins, such as folate (B9) and pantothenate (B5). For the biosynthesis of folate, *B. pseudocatenulatum* JCLA3 contains different genes codifying enzymes, such as para-aminobenzoate synthase glutamine amidotransferase (EC 2.6.1.85), GTP cyclohydrolase IA (EC 3.5.4.16), dihydropteroate synthase (EC 2.5.1.15), dihydroneopterin aldolase (EC 4.1.2.25), 2-amino-4-hydroxy-6-hydroxymethyldihydropteridine diphosphokinase (EC 2.7.6.3), dihydrofolate synthase (EC 6.3.2.12), and folylpolyglutamate synthase (EC6.3.2.17).

In the case of pantothenate, the gene was found codifying to ketol-acid reductoisomerase (EC 1.1.1.86), acetolactate synthase I/III (EC 2.2.1.6), pantetheine-phosphate adenylyltransferase (EC 2.7.7.3), dihydroxy-acid dehydratase (EC 4.2.1.9), holo-[acyl-carrier protein] synthase (EC 2.7.8.7), dephospho-CoA kinase (EC 2.7.1.24), 2-dehydropantoate 2-reductase (EC 1.1.1.169), phosphopantothenoylcysteine decarboxylase (EC 4.1.1.36), phosphopantothenate-cysteine ligase (EC 6.3.2.5), type III pantothenate kinase (EC 2.7.1.33), and branched-chain amino acid aminotransferase (EC 2.6.1.42). Another important biosynthetic capability found in the genome was secondary bile acid biosynthesis by choloylglycine hydrolase (EC 3.5.1.24).

### 3.3. Biochemical Characterization of B. pseudocatenulatum JCLA3

#### 3.3.1. Effect of Carbon Source on Growth

The kinetics of *B. pseudocatenulatum* JCLA3 on different carbon sources are shown in Figure 4. The strain grew in all the substrates and showed similar kinetics (*p* > 0.05), initiating the exponential phase between 4–8 h.

#### 3.3.2. Antibiotic Profile

*B. pseudocatenulatum* JCLA3 displayed resistance to β-lactam antibiotics (ampicillin and cephalothin) as well as *B. animalis* subsp. *lactis* Bb-12. However, *B. animalis* subsp. *lactis* Bb-12 also showed resistance to dicloxacillin and inhibitors of protein and nucleic acid synthesis (Table 1a).

#### 3.3.3. Hemolysis Test

Regarding hemolysis, *B. pseudocatenulatum* JCLA3 and *B. animalis* subsp. *lactis* Bb-12 did not show activity on the blood agar plate (Table 1b).

#### 3.3.4. Antimicrobial Activity

*B. pseudocatenulatum* JCLA3 and *B. animalis* subsp. *lactis* Bb-12 were tested for antimicrobial activity against different strains (Table 1c). The *B. pseudocatenulatum* JCLA3 strain showed a weak inhibitory activity over *E. coli* ATCC 25922 and *S. aureus* ATCC 6538. However, no inhibition was observed for *E. coli* O157:H7 and *S. typhi* ATCC14028. In contrast, the Bb-12 strain showed total inhibition of tested strains except for *S. aureus* ATCC 6538, which was weak.

#### 3.3.5. Bile Salt Tolerance

*B. pseudocatenulatum* JCLA3 and *B. animalis* subsp. *lactis* Bb-12 were able to tolerate concentrations of primary and secondary bile salts, such as the ones found in the GIT (Table 1d). In the cases of secondary bile salts, a halo of precipitation was observed (Figure 5). The genomic analysis showed that *B. pseudocatenulatum* JCLA3 contains the *cbh* gene that codifies to choloylglycine hydrolase (EC 3.5.1.24), which can deconjugate bile salts. In addition, a sodium bile acid symporter family (BASS) was found (data found in the genome).

#### 3.3.6. Tolerance to the Gastrointestinal Tract

The tolerance to sequential simulated gastrointestinal solutions showed that *B. pseudocatenulatum* JCLA3 and *B. animalis* subsp. *lactis* Bb-12 have a 100% viability in SS. *B. pseudocatenulatum* JCLA3 could not survive under SGF due to pH 1.5 and pepsin for 90 min. This pH is commonly present in the stomach during fasting, and the time is similar to the amount of time that food remains in the stomach during digestion. To know which showed the inhibitory effect, we tested pH 1.5 and pepsin independently, but both conditions did not show any growth within 90 min. Then we tested the ability to survive under pH 1.5 at different time periods. As a result, the strain *B. pseudocatenulatum* JCLA3 survived for less than 10 min. In addition, it could not survive under pH 2, 3, and 5 for 90 min. These pH values are like those obtained in the stomach after eating. When the viability was tested non-sequentially in SIF, no viability was found. Nevertheless, when SIF was used without the addition of pancreatin, the loss compared to the initial count was 99.93% (Table 1e).

*B. animalis* subsp. *lactis* Bb-12 tolerated acidity and pepsin with a loss of 99% of viability. When the effect of pH was tested without adding pepsin, the loss diminished to 97%. Additionally, we tested the ability to grow at pH 1.5 during the 90 min period, and it was found that *B. animalis* subsp. *lactis* Bb-12 was 100% viable after 30 min and lost 99.99% of viability at 60 min. When it was grown at pH 2 and 3 after 150 min, a 99% of loss of viability was shown in both cases. The tolerance to SIF was tested non-sequentially in the absence and presence of pancreatin, but no viability was shown.

#### 3.3.7. DPPH Radical Scavenging Activity Assay

*B. pseudocatenulatum* JCLA3 showed antioxidant ability in cell-free extracts and intact cells (5% and 35%, respectively) and *B. animalis* subsp. *lactis* Bb-12 showed 2% and 25%, respectively. In contrast, the cell debris of *B. animalis* subsp. *lactis* Bb-12 demonstrated a higher antioxidant ability (20%) than *B. pseudocatenulatum* JCLA3 (15%) (Figure 6a). Differences among groups were statistically different (*p* < 0.05).

Through genomic annotation, we found that 80 genes could be implicated in the antioxidant potential showed by *B. pseudocatenulatum* JCLA3. Some of these genes are related to the expression of proteins, such as Multifunctional fusion protein (Including Cytidylate kinase (CK) (EC 2.7.4.25), Cytidine monophosphate kinase (CMP kinase), and GTPase Der (GTP-binding protein EngA)), ABC transporter ATP-binding protein, Ferredoxin, and Thioredoxin, among others.

#### 3.3.8. Adhesion Assay

A desirable trait for probiotic bacteria is the ability to adhere to the intestinal epithelium or mucus layer, as this may increase the residence time in the GIT and facilitate interactions with host cells. In this study, we observed that the *B. pseudocatenulatum* JCLA3 strain could adhere up to 4% to the human epithelial cells Caco-2, whereas *B. animalis* subsp. *lactis* Bb-12 increased this property by two-fold (Figure 6) (*p* < 0.05).

Genome analysis of the *B. pseudocatenulatum* JCLA3 strain showed that the presence of *lspA* (Bifido1_01405), *dnaK* (Bifido1_01970), *grpE* (Bifido1_01969), *aprE* (Bifido1_01605), *fimA* (Bifido1_00926), *EF-Tu* (Bifido1_00617), *dppB3* (Bifido1_00550), *grpE* (Bifido1_01968), and *tadE* (Bifido1_00166) genes might be responsible for adhesion.

## 4. Discussion

The genomic and biochemical characterization of microorganisms provides insights into identifying possible human health applications and understanding the mechanism involved. Prior to this study, an isolated bacterium was classified in the genera *Bifidobacterium*; nevertheless, its species was not established. In this study, we classify this microorganism as *B. pseudocatenulatum* using V2-V4 regions and the multilocus analysis, which allowed us to increase the discriminatory power between *Bifidobacterium* species. Few studies have explored the genome of *B. pseudocatenulatum*, and few have related the genome with its biochemical characterization and functionality to establish and understand potential human health applications. In this study, we found that *B. pseudocatenulatum* JCLA3 could metabolize large amounts of mono and oligosaccharides, illustrating the complexity of different carbohydrate consumption and their specific regulation. Bifidobacteria are believed to play an important role in carbohydrate fermentation in the colon. They can indeed ferment various complex carbon sources, such as gastric mucin, xylo-oligosaccharides, (trans)-galactooligosaccharides, soybean oligosaccharides, malto-oligosaccharides, fructo-oligosaccharides, pectin, and other plant derived-oligosaccharides, although the ability to metabolize carbohydrates is species- and strain-dependent and via bifid-shunt such have been reported in *Bifidobacterium* NCC2705 [34].

*B. pseudocatenulatum* JCLA3 contains a variety of genes that code for different enzyme activities (Appendix A), such as glycosyl hydrolases, with the formation of acetate, lactate, ethanol, and even small amounts of succinate, as end products [35], which could reflect its adaptation to the human’s gastrointestinal environment [36]. By genome annotation, we found that *B. pseudocatenulatum* JCLA3 has 44 predicted glycosyl hydrolases (Appendix A), which can act over a wide range of di-, tri-, and higher order oligosaccharides. *Bifidobacterium* carries out an anaerobic fermentative metabolism, obtaining energy in the ATP form by phosphorylation at the substrate level during the hydrolysis of carbohydrates. Glycoside hydrolases are a group of enzymes synthesized by bacteria residing in the colon and used to break down dietary, non-digestible, and plant-derived carbohydrates as an energy source. Once internalized into the cytoplasm, hexose monosaccharides (e.g., fructose and glucose) are converted into acetate and lactate by the fructose 6-phosphate phosphoketolase pathway [37]. Acetate and lactate cross-feeding interact between Bifidobacteria and butyrate-producing colon bacteria in the human colon [38]. In addition to the hydrolases, *B. pseudocatenulatum* JCLA3 contains 70 genes involved in ABC transporter systems, responsible for the uptake of various carbohydrates, six genes related to ATP-binding cassette, and a single PTS (phosphotransferase system). Transport of ribose, stachyose, melibiose, maltose, xylobiose, galactofuranose, and arabinogalactan oligomer/maltooligosaccharide to be facilitated by ABC-type systems in this microorganism, while glucose is internalized using a PTS system. Previously, Olvera et al. (2013) [11] induced gene expression of *ack* and *ldh* in the *Bifidobacterium* studied, which was induced by lactose, inulin, sucrose, and glucose, and reported the mRNA level expression of these genes.

Although *Bifidobacteria* have been studied for over one century, the lack of genetic information has limited insights into their biosynthetic capabilities [39]. *B. pseudocatenulatum* JCLA3 showed a positive association with essential amino acid pathways, such as L-isoleucine biosynthesis I (from threonine), L-valine biosynthesis, and L-lysine biosynthesis VI. The above information suggests that *B. pseudocatenulatum* JCLA3 can be a particularly beneficial bacterium in improving brain function by modulating the availability and metabolism of essential amino acids as a key regulator of the gut-brain axis [10].

Another important biosynthetic capability in the genome was secondary bile acid biosynthesis by choloylglycine hydrolase (EC 3.5.1.24), an enzyme with possible applications in hypercholesterolemia. Some probiotics show hypocholesterolemic effects in animal models due to reductions in lipid and cholesterol levels since BSH can deconjugate bile salts, inducing a de novo synthesis of conjugated salts at the expense of cholesterol, resulting in decreasing serum levels, alterations in energy, homeostasis and the excretion of larger amounts of free bile acids in feces, thus making BSH a clinically significant enzyme [40]. The BSH showed affinity to glycine and taurine secondary conjugates. However, quantification of the activity is required to determine if it is more efficiently deconjugating one or the other [41].

Probiotics have some proposed mechanisms in the elimination of cholesterol, such as the assimilation of cholesterol by growing cells, the binding of cholesterol to the cellular surface, the incorporation of cholesterol into the cellular membrane, the production of short-chain fatty acids by oligosaccharides, the deconjugation of bile via BSH, and the coprecipitation of cholesterol with deconjugated bile [42]. An important aspect shown by the genome of *B. pseudocatenolatum* JCLA3 is the presence of the cholylglycine hydrolase gene (3.5.1.24), an enzyme that catalyzes the hydrolysis of glycine and/or taurine–conjugated bile salts into amino acid residues and free bile acids [43]. Deconjugated bile salts are less soluble and less efficiently reabsorbed from the intestinal lumen than their conjugated counterparts. This results in the excretion of more significant amounts of free bile acids in feces [28].

Several Bifidobacteria are resistant to β-lactam antibiotics, as reported previously [44]. In the case of *B. pseudocatenulatum,* the strain B7003 has shown resistance to ampicillin in higher doses (MIC > 500 mg/mL, 100 μg/mL) [45,46] than those tested in this study. However, the resistance depends on the method used and the unrelatedness of the strains; then, there is still a lack of agreement in the resistance susceptibility breakpoints for most antibiotics [47]. Di Gioia et al. (2013) [46] evaluated the presence of *blaCTX-M1* and two genes that codify to β-lactamases, related to β-lactam resistance. Nevertheless, we did not find these, but at least ten genes were found to be related to β-lactam resistance. These genes, in general, codify to peptidoglycan synthase, putative ABC transporter substrate-binding or ATP-binding component, putative β-hexosaminidase, dipeptide ABC transporter ATP-binding, permease, or substrate-binding components. Particularly, we found *oppA*, *oppB*, *oppC*, and *oppF* genes are involved in Quorum sensing (Qs) and β-lactam resistance; as has been reported, the signals of the Qs could trigger changes in gene expression, regulate various cellular processes, which mainly involve the regulation of drug resistance and other. *B. pseudocatenulatum* JCLA3 does not contain plasmids, which represents an important feature considering that transmissible antibiotic resistances are in most strains encoded by plasmid DNA. Therefore, *B. pseudocatenulatum* JCLA3 possesses resistance to some antibiotics, but they are all chromosomally encoded. Therefore, their spread to other bacteria can be considered a rare event [46].

Regarding hemolysis, *B. pseudocatenulatum* JCLA3 and *B. animalis* subsp. *lactis* Bb-12 did not show activity on the blood agar plate (Table 1b), in agreement with other studies showing that this activity is rarely present in *Bifidobacterium* [48].

*B. pseudocatenulatum* JCLA3 and *B. animalis* subsp. *lactis* Bb-12 showed different levels of antimicrobial activity probably due to different mechanisms, such as the production of short-chain fatty acids, mainly acetic and lactic acids, the production of hydrogen peroxide, which is the mechanism used to inhibit *S. aureus*, and the production of proteinaceous compounds, such as bacteriocins [49,50,51].

The tolerance to sequential simulated gastrointestinal solutions showed *that B. pseudocatenulatum* JCLA3 could overcome simulated saliva, so we hypothesized that the time that food is in the mouth is not enough for lysozymes to produce damage in the *B. pseudocatenulatum* JCLA3 cell structure. However, the conditions found in the stomach and intestine during fasting or digestion can be very aggressive and, in some cases, can have a synergistic effect on the cell wall structure, such as in the case of the mix of bile salts used in the SIF since *B. pseudocatenulatum* JCLA3 was able to grow in the presence of primary bile salts (Table 1e). Finally, these results indicate that if *B. pseudocatenulatum* JCLA3 will, in turn, be used as a supplement, it must be administered by using a protective matrix. In the case of *B. animalis* subsp. *lactis* Bb-12 (control strain) acidity and the mix of bile salts strongly affected viability. It has been well known that the viability of Bifidobacteria in gastric and intestinal juices is different at the strain level and is affected by several factors. These factors include the degree of acidity in the stomach, the duration of exposure to acids in the gastric juice, the concentration and duration of exposure to bile salts, the level of BSH activity, the enzymes present in the gastrointestinal tract, intestinal motility, as well as the bacterial phase growth and to the method of delivery of the bacteria to the digestive tract [52].

Whether Bifidobacteria colonize from birth or are ingested as a probiotic, they will encounter and must overcome stresses in the GIT with bile being the major stress-inducing factor to bacteria, due to its bactericidal properties. Different mechanisms of bile resistance by Bifidobacteria have been set, including the efflux of bile salts by multi-drug transporters, compositional changes of the cell membrane, F_0_F_1_-ATPase proton efflux, changes in metabolism, and hydrolysis of bile salts [53]. Previous studies have shown that biofilm formation by *Bifidobacterium* protects them from high concentrations of bile. In the genome of *B. pseudocatenulatum* JCLA3, we found genes related to the production of biofilm components, which codify to glycan, exopolysaccharide (EPS), and lipopolysaccharide biosynthesis, many glycosyltransferases related to exopolysaccharide biosynthesis [54] and an AI-2 transporter was found. In fact, AI-2-dependent quorum sensing has been implicated in the formation of biofilm in other bacteria [55]. *B. pseudocatenulatum* JCLA3 is one of the Bifidobacteria members able to produce extracellular EPS [54]. Nevertheless, it is necessary to test in vitro if *B. pseudocatenulatum* JCLA3 can produce EPS and biofilm, and to investigate the specific in vitro conditions that allow their production. In addition, it will be necessary to characterize them since, for instance, EPS with high molecular weight has been able to suppress the production of pro-inflammatory cytokines. In contrast, EPS with small molecular weight or acidic EPS had immune-stimulating properties in *Bifidobacterium* [54]. In addition, the response to high bile concentration in *Bifidobacterium* has been shown to involve a specific response in carbohydrate metabolism. It has been reported that fatty acid synthesis is important for bile resistance when Bifidobacteria are exposed to bile; therefore, changes in surface hydrophobicity and perhaps membrane permeability due to altered fatty acid synthesis may help to resist the bactericidal effects of bile [53]. In the case of *B. animalis* subsp. *lactis* Bb-12, its growth in bile salts, and the presence of the gene coding for BSH have been previously reported [56].

Reactive oxygen species (ROS) are produced by aerobic respiration and the immune defense of organisms. Excessive amounts of ROS can result in cellular damage, which promotes chronic diseases, such as cardiovascular diseases, diabetes, and cancer. Certain probiotic strains present significant antioxidant effects and can act as antioxidants to maintain intestinal redox balance in the gut by adhering to and colonizing the intestinal lumen [57].

The antioxidative potential of Bifidobacteria has been previously reported using the DPPH free radical scavenging method. We reported that intact cells and cell-free extracts of *B. pseudocatenulatum* JCLA3 showed higher ability in vitro, which was similar to the antioxidant activity of Bifidobacteria isolated described by Shen et al. (2011) [58]; Kim et al. (2003) [59] and Lin and Chang, (2000). The antioxidative effects in *B. pseudocatenulatum* JCLA3 could have been present because Bifidobacteria can produce antioxidative compounds [60], and polymers with antioxidant protective effects such as EPS [61]. In addition, we observed in *B. pseudocatenulatum* JCLA3 antioxidant-related genes, such as *PNPOx* (pyridoxine 5′-phosphate oxidase); *AhpC* (alkyl hydroperoxide reductase subunit C) considered as an antioxidant protein; *Bcp* (tiol peroxidase) a member of peroxiredoxin, which exhibits hydroperoxide peroxidase activities; *trxA* (thioredoxin), *trxB* (thioredoxin reductase) thioredoxin system plays a crucial role in defense against ROS for anaerobes, and *nrdH* (glutaredoxin), and *mntH* (manganese transport protein), a transporter selective for manganese that acts as a co-factor of antioxidant enzymes and non-proteinaceous manganese antioxidants [57]. In addition, we found that *trxB* and *AhpC* and, on the other hand, *nrdH* and *nrdl* (ribonucleotide reductase stimulatory protein) were found as operons.

The adhesion of probiotics to human GIT Is considered one of the main criteria for the bacteria to benefit health. The adhesion ability of probiotics is dependent and is influenced by physicochemical and biological conditions. It depends on environmental factors, such as pH and temperature, the ionic strength of the medium, as well as characteristics of both the bacteria and substrate, such as surface free energy, the presence of bacterial surface structures, such as specific attachment proteins, and their hydrophobicity or hydrophilicity. The genome of the *B. pseudocatenulatum* JCLA3 strain contains cell adhesion genes associated with probiotic properties: fibronectin/fibrinogen binding proteins and mucus-binding proteins. As reported by Westermean et al. [62], *Bifidobacterium* species have reported the presence of moonlighting proteins involved in the adhesion of bacteria to host tissues [62], such as transaldolases (mucin binding protein), enolase, glyceraldehyde-3-phosphate, DnaK, BSH, and phosphoglycerate mutase among others).

## 5. Conclusions

The complete genome of *B. pseudocatenulatum* JCLA3 was comparable in size to other *B. pseudocatenulatum* strains. Its knowledge allows us to further understand the role of this bacterium in the metabolism of carbohydrates and the catalytic tools it uses to adapt in the gastrointestinal tract of mammals. It also provides valuable information on the details of its possible functional properties, such as antimicrobial, BSH, and antioxidant activity, adhesion, and certain safety at the genotype level and, in turn, will increase the knowledge of its possible properties and the genes related to possible health effects and may lead to genome-based biotechnological applications in human healthcare and food science.

## Figures and Tables

**Figure 1 microorganisms-10-02100-f001:**
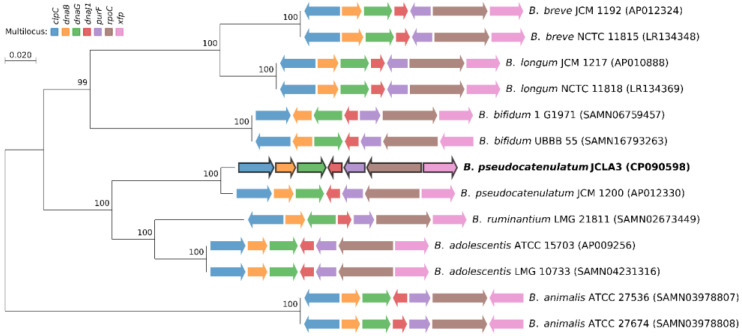
Multilocus sequence analysis and Maximum Likelihood phylogenetic tree based on bifidobacterial genomes sequences available in GenBank. *B. pseudocatenulatum* JCLA3 identified and characterized in this study are denoted in bold type. The solid arrows indicate the genes used for the construction of the multilocus, and the direction reflects their orientation within the genome of each Bifidobacterium strain.

**Figure 2 microorganisms-10-02100-f002:**
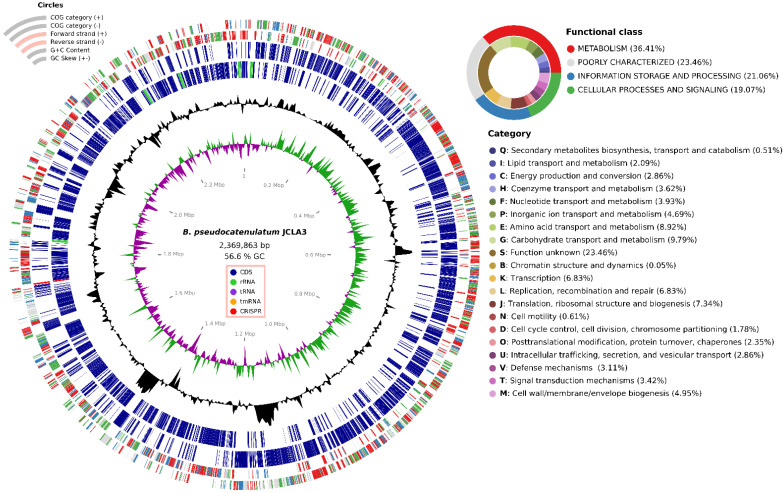
Circular genomic map of *B. pseudocatenulatum* JCLA3. Genes were grouped into categories according to their functionality.

**Figure 3 microorganisms-10-02100-f003:**
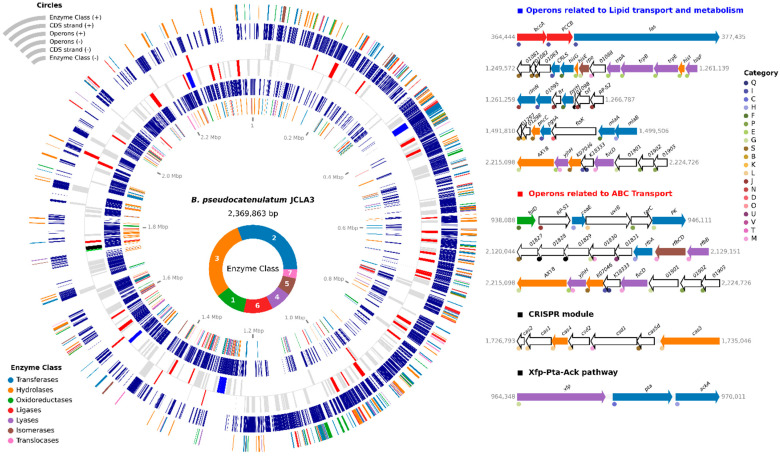
Grouped enzymes according to their functionality and some operons coding on the *B. pseudocatenulatum* JCLA3 genome.

**Figure 4 microorganisms-10-02100-f004:**
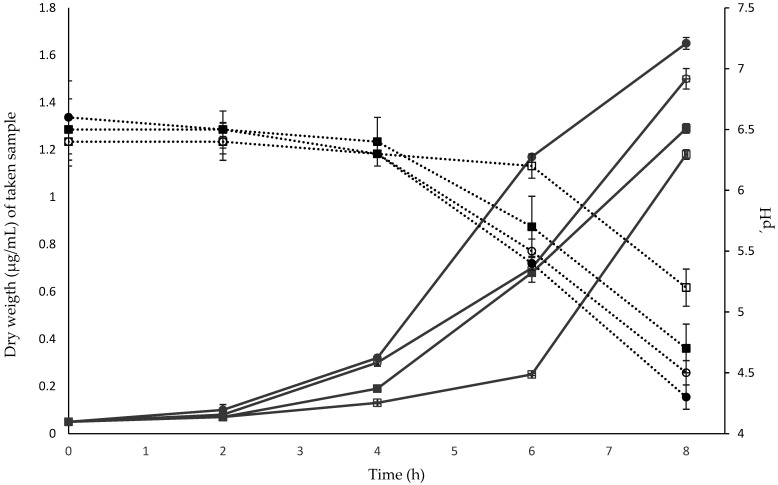
Effect of different carbon sources on the growth of *B. pseudocatenulatum* JCLA3. Continuous lines in grey correspond to the biomass obtained, and dotted lines in black indicate changes in pH regarding different carbon sources. ○ glucose (control); ● sucrose; ■ inulin; □ lactose; and pH of the medium along time.

**Figure 5 microorganisms-10-02100-f005:**
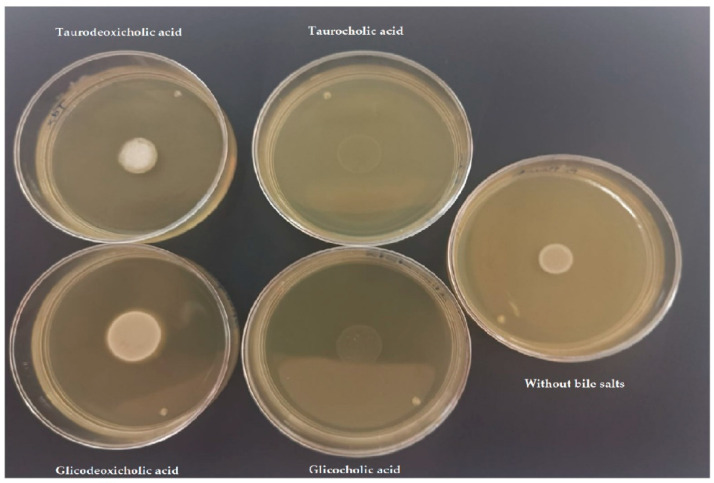
Bile salt tolerance. Growth of one colony of *B. pseudocatenulatum* JCLA3 in TPY medium added with glico and tauro cholic acids (primary bile salts) and glico and taurodeoxycholic acids (secondary bile salts) at 0.5%.

**Figure 6 microorganisms-10-02100-f006:**
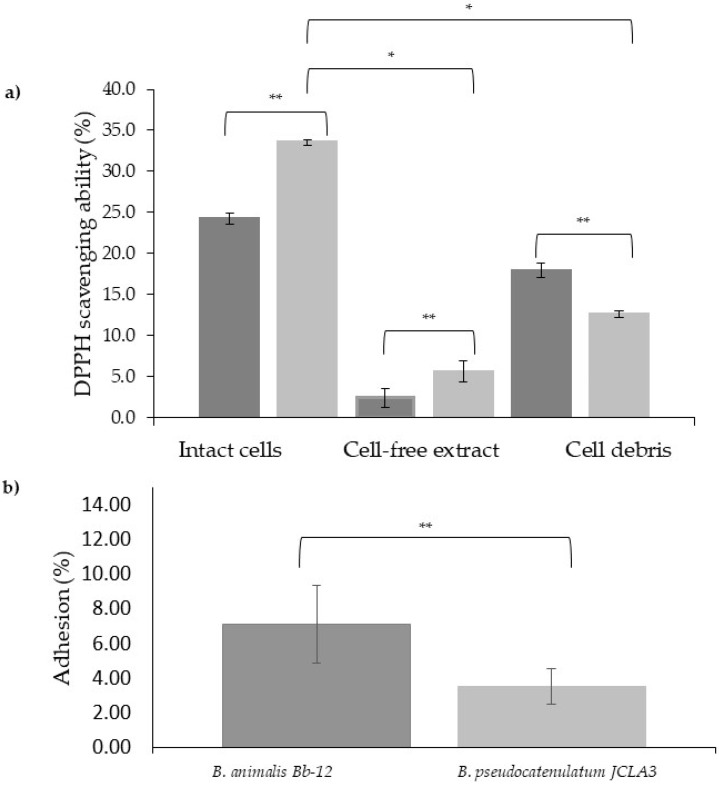
(**a**) DPPH scavenging ability in intact cells, cell-free extract, and cell debris to ■
*B. animalis* subsp. *lactis* Bb-12 and ■ *B. pseudocatenulatum* JCLA3. (**b**) Percentage of adhesion ■ *B. animalis* subsp. *lactis* Bb-12 and ■ *B. pseudocatenulatum* JCLA3. * Means significant difference between the same microorganism but different sample. ** means significant difference between different Bifidobacterium species.

**Table 1 microorganisms-10-02100-t001:** Biochemical characterization of *B. pseudocatenulatum* JCLA3.

		*Strains*
	*B. pseudocatenulatum* JCLA3	*B. animalis* subsp. *lactis* Bb-12
(a) Antibiotic profile * Vancomycin	S	S
Ampicillin	R	R
Trimethoprim sulfamethoxazole	S	S
Gentamicin	S	S
Dicloxacillin	S	R
Cephalothin	R	R
Clindamycin	S	S
Erythromycin	S	S
Penicillin	S	S
Tetracycline	S	R
Cefotaxime	S	S
Ciprofloxacin	S	R
(b) Hemolysis	−	−
(c) Antimicrobial activity		
*E. coli* ATCC 25922	+	++
*Escherichia coli* O157: H7	−	++
*Salmonella typhi* ATCC14028	−	++
*Staphylococcus aureus* ATCC 6538	+	+
(d) Bile salt tolerance assay		
Glycocholic acid 0.1–0.5%	+	+
Taurocholic acid 0.1–0.5%	+	+
Glycodeoxycholic acid 0.1–0.5%	+	+
Taurodeoxycholic acid 0.1–0.5%	+	+
(e) Tolerance to GIT		
Simulated saliva	100%	100%
Simulated gastric fluid	0%	2.26%
Simulated intestinal fluid	0%	0%

* Antibiotic test: S: sensible; R: resistance; +: growth in the medium containing each bile salt. Antimicrobial activity: no inhibition (−), moderate (+), and total inhibition (++).

## Data Availability

The data presented in this study are openly available from the NCBI in the BioProject PRJNA795522.

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
