# Peer review of "Genomic and Biochemical Characterization of Bifidobacterium pseudocatenulatum JCLA3 Isolated from Human Intestine"

_microorganisms, 2022, doi:10.3390/microorganisms10112100_

Round 1

Reviewer 1 Report

This study by González-Vázquez analyzed the genome sequence of Bifidobacterium pesudocatenulatum JCLA3 from the human intestine and used a series of biochemical tests to investigate the correlation between genotype and phenotype in the same strain. This reviewer suggests the following modifications/improvements before this manuscript can be accepted for publication.

1) Introduction section needs more description of the logic and importance of this study. What we can exactly obtain from the bacterial genome, Bifidobacterium, and especially the strain used in this study.

2) The connection between the biochemical characterization (phenotype) and the genome DNA information (genotype) is still weak. One solution is that the authors can start from the results and conclusions of comparison of the JCLA3 strain with the inter-species available in the database, and the one authors used in the manuscript. So some re-arrangement of the results part seems required. Additionally, more comparison analysis should be done to assign more significance to the strain used in this study.

3) Fig. 4: the right Y-axis represents pH from dotted lines. So the continuous lines should not be matched to the pH. The X-axis can be extended a little bit (e.g., 8.5). From this figure, JCLA3 seems to use those different carbon sources equally as seen from 4h to 8h? Is this usual in Bifidobacterium? How about the final concentration in terms of the C number? As for the selection of the carbon source tested for JCLA3, did the author also investigate Bb-12 as a "control"? 

4) Fig. 5A and 5B are not clear. The full plate should be shown here.

5) Fig. 6: Re-arrangement of labeling is required. Statistical analysis with P-value should be performed among samples. 

Author Response

  • Introduction section needs more description of the logic and importance of this study. What we can exactly obtain from the bacterial genome, Bifidobacterium, and especially the strain used in this study.

Answer:

The introduction was modified, highlighting the logic and importance of studying the probiotic characteristics of Bifidobacterium and Bifidobacterium pseudocatenulatum and underscoring the importance of its genome and biochemical characterization.

  • The connection between the biochemical characterization (phenotype) and the genome DNA information (genotype) is still weak. One solution is that the authors can start from the results and conclusions of comparison of the JCLA3 strain with the inter-species available in the database, and the one authors used in the manuscript. So some re-arrangement of the results part seems required. Additionally, more comparison analysis should be done to assign more significance to the strain used in this study.

Answer:

It was done an analysis of the section where the relation between biochemical data and genome information was weak, we found that antioxidant activity and adhesion did not show an association relation. A genomic comparison between B. pseudocatenulatum JCLA3 and other genomes of the same microorganism available in the database was used to find genes related to the biochemical activities tested in this study.

Antioxidant activity and adhesion results regarding genome analysis were included. In addition, a discussion about these results was included.

  • 4: the right Y-axis represents pH from dotted lines. So, the continuous lines should not be matched to the pH. The X-axis can be extended a little bit (e.g., 8.5). From this figure, JCLA3 seems to use those different carbon sources equally as seen from 4h to 8h? Is this usual in Bifidobacterium? How about the final concentration in terms of the C number? As for the selection of the carbon source tested for JCLA3, did the author also investigate Bb-12 as a "control"?

Answer:

Figure 4 was modified. The length of the x-axis was modified to 8.5, and SD was included in all points of growth and pH.

By the statistical analysis, we found that B. pseudocatenulatum JCLA3 uses the same way different carbon sources (p>0.05); this is usual for Bifidobacterium. We tested the same carbon sources with B. animalis BB-12 and found the same tendency. Nevertheless, we did not include this result in the study since we just used these data as an internal control.

We did not determine the CFU in the experiment since we followed the growth by absorbance. Then we use a standard curve previously performed to correlate dry weight with absorbance, which was indicated in the methodology section.

  • 5A and 5B are not clear. The full plate should be shown here.

Answer:

The experiment was done again, including primary and secondary bile salts, and results were included. Table1 and figure 5 were modified and now figure 5 shows full plates.

  • 6: Re-arrangement of labeling is required. Statistical analysis with P-value should be performed among samples. 

Answer:

The statistical analysis section was included in the methodology. Along results, the significance was included. Figure 6 labeling was modified.

Reviewer 2 Report

This manuscript is very interesting and in my opinion should be published. The subject matter is well known and has been published many times, however, analyses on any new potentially probiotic strain are worth publishing.

In addition, the Authors should clarify the convergence of some analyses with the data published in the article „Characterisation of a Bifidobacterium sp. strain isolated from human faeces and its expression” written by Olvera Rangel N., Gutiérrez Nava A., Azaola Espinosa A., Mayorga Reyes L. (African Journal of Microbiology Research 2013). In this context, the Authors must also identify aspects of the novelty of the results published in the current manuscript.

In addition, the Authors need to improve this text and add missing data. In this regard, my suggestions are as follows:

·         Please correct the spelling of Latin bacterial names throughout the text - they must be written in italics.

·         Please use the correct and complete spelling of the species name of the strain Bifidobacterium animalis subsp. lactis Bb-12 throughout the text.

·         Please explain what the abbreviation BSH means when it is first used in this manuscript.

·         Please discuss the methodological scope of this research and the results obtained with the publication „Characterisation of a Bifidobacterium sp. strain isolated from human faeces and its expression” written by Olvera Rangel N., Gutiérrez Nava A., Azaola Espinosa A., Mayorga Reyes L. (African Journal of Microbiology Research 2013)

·         Section „2. Materials and Methods” – please add information about the statistical analysis of the results performed.

·         Section „2.5. Effect of carbon source ongrowth of Bifidobacterium pseudocatenulatum JCLA3” – please add information about the temperature conditions of the experiment.

·         Section „2.7. Hemolysis test” – please explain why a different medium was used than in the sections „2.5. Effect of carbon source ongrowth of Bifidobacterium pseudocatenulatum JCLA3”, „2.8. Antimicrobial Activity” and “2.9. Bile salt tolerance assay”. Please add information about the conditions under which the bacteria are cultured.

·         Section „2.11. DPPH radical scavenging activity assay” – Please state the exact conditions under which the experiment was conducted. Please complete the information on the number of times the experiment was performed.

·         Lines 443-447 – please discuss this topic with DOI: 10.1016/j.foodres.2004.08.003, DOI: 10.5772/intechopen.88462, doi: 10.1046/j.1365-2672.2000.00792.x and doi: 10.1159/000354316.

·         Lines 474-483 - please discuss this topic with doi: 10.3402/mehd.v26.27812

·         Figure 4 – please add SD values or results of statistical analysis of the results.

·         Please correct Figure 5a i b – the figures are too small and difficult to read.

·         Please correct Figure 5c – the deconjugation reaction is catalysed by the enzyme BSH, please indicate this on the figure.

Author Response

Reviewer 2

  • In addition, the Authors should clarify the convergence of some analyses with the data published in the article „Characterisation of a Bifidobacterium sp. strain isolated from human faeces and its expression” written by Olvera Rangel N., Gutiérrez Nava A., Azaola Espinosa A., Mayorga Reyes L. (African Journal of Microbiology Research 2013). In this context, the Authors must also identify aspects of the novelty of the results published in the current manuscript.

Answer:

We clarify in the introduction the convergence of the study of Olvera et al. (2013) and the difference with ours. The novelty of our paper is the genomic and biochemical characterization of the potential probiotic of a strain isolated from a Mexican natural birth newborn, breastfeeding, that has not been previously published anywhere. Also, genomic analysis done with other reported genomes of the same microorganism showed differences between the same species.

  • In addition, the Authors need to improve this text and add missing data. In this regard, my suggestions are as follows:

Answer:

We eliminated “data not show” because the data were not relevant.

  • Please correct the spelling of Latin bacterial names throughout the text - they must be written in italics.

Answer:

The name of all bacteria along the document was corrected and written in italics.

  • Please use the correct and complete spelling of the species name of the strain Bifidobacterium animalis subsp. lactis Bb-12 throughout the text.

Answer:

The name of B. animalis subsp. lactis Bb-12 was modified along the text.

  • Please explain what the abbreviation BSH means when it is first used in this manuscript.

Answer:

BSH abbreviation was explained in the introduction when it was used for the first time.

  • Please discuss the methodological scope of this research and the results obtained with the publication „Characterisation of a Bifidobacterium sp. strain isolated from human faeces and its expression” written by Olvera Rangel N., Gutiérrez Nava A., Azaola Espinosa A., Mayorga Reyes L. (African Journal of Microbiology Research 2013)

Answer:

We clarify in the introduction the convergence of the study of Olvera et al. (2013) and the difference with ours. The novelty of our paper is the genomic and biochemical characterization of the potential probiotic of a strain isolated from a Mexican natural birth newborn, breastfeeding, that has not been previously published anywhere. Also, genomic analysis done with other reported genomes of the same microorganism showed differences between the same species.

  • Section „2. Materials and Methods” – please add information about the statistical analysis of the results performed.

Answer:

The statistical analysis section on materials and methods was included, and significant differences were included in the results.

  • Section „2.5. Effect of carbon source ongrowth of Bifidobacterium pseudocatenulatum JCLA3” – please add information about the temperature conditions of the experiment.

Answer:

The temperature was included in the methodology

  • Section „2.7. Hemolysis test” – please explain why a different medium was used than in the sections „2.5. Effect of carbon source on growth of Bifidobacterium pseudocatenulatum JCLA3”, „2.8. Antimicrobial Activity” and “2.9. Bile salt tolerance assay”.

Please add information about the conditions under which the bacteria are cultured.

Answer:

To carry out the hemolysis test, we used the medium reported by González-Vázquez et al. (2015) [1]. On the other hand, TPY medium is usually used to grow Bifidobacterium [2-5].

Details about the conditions of the different experiments were included in the methodology section.

  • Section „2.11. DPPH radical scavenging activity assay” – Please state the exact conditions under which the experiment was conducted. Please complete the information on the number of times the experiment was performed.

Answer:

Details about the methodology of DPPH used were included.

  • Lines 443-447 – please discuss this topic with DOI: 10.1016/j.foodres.2004.08.003, DOI: 10.5772/intechopen.88462, doi: 10.1046/j.1365-2672.2000.00792.x and doi: 10.1159/000354316.

Answer:

The references suggested by the reviewer were considered in the discussion of bile salt hydrolase activity. 

  • Lines 474-483 - please discuss this topic with doi: 10.3402/mehd.v26.27812

Answer:

Lines 474-483 were modified using the reference suggested.

  • Figure 4 – please add SD values or results of statistical analysis of the results.

Answer:

SD data were included in figure 4

  • Please correct Figure 5a i b – the figures are too small and difficult to read.

Answer:

The experiment was done again, including primary and secondary bile salts, and the results were included. Table 1 and figure 5 were modified; now, figure 5 show full plates.

  • Please correct Figure 5c – the deconjugation reaction is catalyzed by the enzyme BSH, please indicate this on the figure.

Answer:

The reaction was eliminated due to its poor significance since previously reported.

  1. González-Vázquez, R.; Azaola-Espinosa, A.; Mayorga-Reyes, L.; Reyes-Nava, L.A.; Shah, N.P.; Rivera-Espinoza, Y. Isolation, Identification and Partial Characterization of a Lactobacillus casei Strain with Bile Salt Hydrolase Activity from Pulque. Probiotics Antimicrob Proteins 2015, 7, 242-248, doi:10.1007/s12602-015-9202-x.
  2. GonzÁLez-SÁNchez, F.; Azaola, A.; GutiÉRrez-LÓPez, G.F.; HernÁNdez-SÁNchez, H. Viability of microencapsulated Bifidobacterium animalis ssp. lactis BB12 in kefir during refrigerated storage. International Journal of Dairy Technology 2010, 63, 431-436, doi:https://doi.org/10.1111/j.1471-0307.2010.00604.x.
  3. Mayorga-Reyes, L.; Bustamante-Camilo, P.; Gutiérrez-Nava, A.; Barranco-Florido, E.; Azaola-Espinosa, A. CRECIMIENTO, SOBREVIVENCIA Y ADAPTACIÓN DE Bifidobacterium infantis A CONDICIONES ÁCIDAS GROWTH, SURVIVAL AND ADAPTATION OF Bifidobacterium infantis TO ACIDIC CONDITIONS.
  4. González, R.; Blancas, A.; Santillana, R.; Azaola, A.; Wacher, C. Growth and final product formation by Bifidobacterium infantis in aerated fermentations. Applied Microbiology and Biotechnology 2004, 65, 606-610, doi:10.1007/s00253-004-1603-9.
  5. Grill, J.P.; Cayuela, C.; Antoine, J.M.; Schneider, F. Effects of Lactobacillus amylovorus and Bifidobacterium breve on cholesterol. Letters in Applied Microbiology 2000, 31, 154-156, doi:https://doi.org/10.1046/j.1365-2672.2000.00792.x.

Reviewer 3 Report

In this manuscript the authors claim to resolve the genomic and biochemical characteristics of Bifidobacterium JCLA3. The benefits that Bifidobacterium bifidum brings to humans are well known, even empathetic. We should encourage more research on bifidobacteria. However, this manuscript lacks some description and highlighting of the novelty of JCLA3, especially in the abstract section. As we all know, we need innovative research, not duplicative research. The manuscript should be more descriptive of the novelty of the origin/genomic/biochemical features of JCLA3, in short, we need innovative studies, even if a little.

     There is not much wrong with the details, but what exactly the spacing symbols should be between keywords should be carefully confirmed.

Author Response

Reviewer 3

  • In this manuscript the authors claim to resolve the genomic and biochemical characteristics of Bifidobacterium  The benefits that Bifidobacterium bifidum brings to humans are well known, even empathetic. We should encourage more research on bifidobacteria. However, this manuscript lacks some description and highlighting of the novelty of JCLA3, especially in the abstract section. As we all know, we need innovative research, not duplicative research. The manuscript should be more descriptive of the novelty of the origin/genomic/biochemical features of JCLA3, in short, we need innovative studies, even if a little.

Answer:

In this work, we studied a strain of B. pseudocatenulatum isolated in Mexico from a natural birth newborn and breastfeeding. We determined some biochemical characteristics and the complete genome, which was analyzed and compared with the genomes of this genus and specie previously reported. It is worth noting that differences in the genomes were found regarding the other reported.

When a new potential probiotic strain is isolated to know its potential, a characterization is required, which includes many of the mythologies used in the present study, we consider that without this “basal” characterization, poor knowledge about its security and functionality will be stablished, and as have been recommended by organizations such as OMS, EFSA, and other, probiotics have to be characterized since the most basic until its novel functionality. Nevertheless, our characterization is deep since the genome information obtained has led to establishing, for instance, that despite of the strain showing antibiotic resistance, this resistance is chromosomally, and no risk in terms of genomic transference could be possible.

The novel functionality of this strain is being determined at this time in our laboratory. However, the results are considered for a further paper.

Round 2

Reviewer 1 Report

This reviewer has no further comments on the revised manuscript. Two minor suggestions are 1) the authors can provide the "Data not shown" as supplemental results; 2) if P<0.5, also mark * to figures.

Author Response

Moderate English changes required

Answer:

All the author reviewed the English of the paper to modify mistakes.

This reviewer has no further comments on the revised manuscript. Two minor suggestions are 1) the authors can provide the "Data not shown" as supplemental results; 2) if P<0.5, also mark * to figures.

Answers:

  • The experiments in which we used the legend “data not shown” where the ones associated to the effect of pH over viability. In this case we observed that after 90 min of evaluation under pH 2,3 and 5 no viability was detected. If we draft this result, the same graph will be for all the pH since the UFC count decrease drastically from 1 x 108 UFC/mL to almost zero UFC. Finally, we consider that this information should not be in supplemental results.
  • A mark indicating significant differences was added in figure 6. Also, the legend of figure 6 was modified indicating the significance of asterisks.

Reviewer 2 Report

I see that the Authors have improved the manuscript taking into account the reviewers' suggestions. I have no further comments on this manuscript. I consent to its publication in the current version. 

Author Response

Comments to Reviewers

Reviewer 2

English language and style are fine/minor spell check required.

Answer:

All the author reviewed the English of the paper to modify mistakes.

I see that the Authors have improved the manuscript taking into account the reviewers' suggestions. I have no further comments on this manuscript. I consent to its publication in the current version. 

Reviewer 3 Report

The authors have made a number of changes to the manuscript which have resulted in a good improvement in its quality. However, the issue of novelty, which I am most concerned about, is still more than not prominent, but it is barely acceptable. In addition, the spacing symbols between keywords in the revised version are inconsistent, but in fact they have not been changed. Why?

Author Response

Reviewer 3

The authors have made a number of changes to the manuscript which have resulted in a good improvement in its quality. However, the issue of novelty, which I am most concerned about, is still more than not prominent, but it is barely acceptable. In addition, the spacing symbols between keywords in the revised version are inconsistent, but in fact they have not been changed Why?

Answer:

There are few genomic and biochemical reports about B. pseudocatenulatum  in the literature and any about the isolation and genomic-biochemical characterization of this strain, isolated from Mexican newborns, breastfeed. Regarding the genome, we are working in a genome comparison of B. pseudocatenulatum strains using a novel strategy. In addition, there are few reports about qualitatively bile salt hydrolase activity in B. pseudocatenulatum. In this study we are considering several parameters that a probiotic strain should show, since we think that a previous characterization must exist before proving a beneficial effect.

On the other hand, spacing symbols were modified according to the reviewer´s comment.
